# Selection Effects and COVID-19 Mortality Risk after Pfizer vs. Moderna Vaccination: Evidence from Linked Mortality and Vaccination Records

**DOI:** 10.3390/vaccines11050971

**Published:** 2023-05-11

**Authors:** Vladimir Atanasov, Natalia Barreto, Jeff Whittle, John Meurer, Benjamin W. Weston, Qian (Eric) Luo, Andy Ye Yuan, Lorenzo Franchi, Ruohao Zhang, Bernard Black

**Affiliations:** 1Mason College of Business, William & Mary, Williamsburg, VA 23185, USA; vladimir.atanasov@mason.wm.edu; 2Department of Economics, University of Illinois, Urbana-Champaign, Champaign, IL 61820, USA; barretoparrapaulanatalia@gmail.com; 3Medical College of Wisconsin; Milwaukee, WI 53226, USA; jwhittle@wi.rr.com (J.W.); jmeurer@mcw.edu (J.M.); beweston@mcw.edu (B.W.W.); 4Department of Health Policy and Management, George Washington University, Washington, DC 20052, USA; qluo@gwu.edu; 5Pritzker School of Law, Northwestern University, Chicago, IL 60611, USA; 6Department of Agricultural Economics, Pennsylvania State University, State College, PA 16803, USA

**Keywords:** COVID-19 mortality rates, COVID-19 excess mortality percentage, vaccine effectiveness, Moderna vaccine, mRNA1273, Pfizer vaccine, Pfizer-BioNTech vaccine, BNT162b2

## Abstract

Prior research generally finds that the Pfizer-BioNTech (BNT162b2) and Moderna (mRNA1273) COVID-19 vaccines provide similar protection against mortality, sometimes with a Moderna advantage due to slower waning. However, most comparisons do not address selection effects for those who are vaccinated and with which vaccine. We report evidence on large selection effects, and use a novel method to control for these effects. Instead of directly studying COVID-19 mortality, we study the COVID-19 excess mortality percentage (CEMP), defined as the COVID-19 deaths divided by non-COVID-19 natural deaths for the same population, converted to a percentage. The CEMP measure uses non-COVID-19 natural deaths to proxy for population health and control for selection effects. We report the relative mortality risk (RMR) for each vaccine relative to the unvaccinated population and to the other vaccine, using linked mortality and vaccination records for all adults in Milwaukee County, Wisconsin, from 1 April 2021 through 30 June 2022. For two-dose vaccinees aged 60+, RMRs for Pfizer vaccinees were consistently over twice those for Moderna, and averaged 248% of Moderna (95% CI = 175%,353%). In the Omicron period, Pfizer RMR was 57% versus 23% for Moderna. Both vaccines demonstrated waning of two-dose effectiveness over time, especially for ages 60+. For booster recipients, the Pfizer–Moderna gap is much smaller and statistically insignificant. A possible explanation for the Moderna advantage for older persons is the higher Moderna dose of 100 μg, versus 30 μg for Pfizer. Younger persons (aged 18–59) were well-protected against death by two doses of either vaccine, and highly protected by three doses (no deaths among over 100,000 vaccinees). These results support the importance of a booster dose for ages 60+, especially for Pfizer recipients. They suggest, but do not prove, that a larger vaccine dose may be appropriate for older persons than for younger persons.

## 1. Introduction

COVID-19 vaccines have saved hundreds of thousands of lives in the United States (Steele 2022) and millions worldwide (Watson 2022). Conventional wisdom is that both mRNA vaccines, from Pfizer-BioNTech (BNT162b2) and Moderna (mRNA1273), provide strong and similar protection against severe disease and mortality. Public health guidelines on vaccination recommend both vaccines equally [1,2,3]. Prior research comparing the two vaccines, reviewed in the Discussion Section, is limited and mixed, but suggests a nod to Moderna because it wanes more slowly. 

However, these studies rely on observational data, and are vulnerable to selection effects for those who are vaccinated, with how many doses, and with which vaccine. Failure to address these selection effects can lead to biased estimates of vaccine effectiveness (VE), relative mortality risk (RMR) for the vaccinated relative to the unvaccinated, and RMR for Pfizer versus Moderna vaccinees [4]. In prior work, we provide evidence for strong selection effects on which persons are vaccinated; two-dose vaccinees are healthier than the unvaccinated, and three-dose vaccinees are generally healthier than two-dose vaccinees [5]. Here, we extend that work and provide evidence for important selection effects in which vaccine people receive. We then study RMR for each vaccine versus the unvaccinated, and compare the two mRNA vaccines, controlling for these selection effects. 

Many studies have reported real-world evidence on vaccine effectiveness (VE) for the mRNA vaccines against infection, hospitalization, and death (for brevity, we cite principally systematic reviews) [6,7,8,9]. However, many studies report results for both vaccines together [5,10,11,12,13,14,15]. Most studies that report vaccine-specific results have limited controls for individual characteristics, often only age and gender; use research designs that are prone to selection effects, often a test-negative design, which compares people with positive COVID-19 tests to controls who test negative; and/or cover limited time periods or populations (e.g., US veterans). After controlling for selection effects, and in contrast to the mixed results from these studies, we find evidence of a large mortality-reduction advantage for Moderna over Pfizer for two-dose vaccinees, aged 60+. Both vaccines provide similar protection against mortality for younger ages, and after a booster dose.

Both vaccines use similar mRNA technology, but the initial two Moderna doses were 100 μg versus 30 μg for Pfizer (the booster doses are 50 μg for Moderna versus 30 μg for Pfizer). The differing doses reflect each company’s initial decision on how to balance the extra immune-system boost from a larger dose versus the increased risk of side effects. Yet, we know from research on vaccines for other respiratory diseases (influenza and pneumonia) that VE declines with age among the elderly [16,17]. There is also evidence for influenza that the elderly benefit from a larger vaccine dose that compensates for their weaker immune systems [16,17,18,19,20]. This pattern could plausibly hold for the COVID-19 vaccines, and could explain the observed Moderna advantage for ages 60+.

In prior work, we propose, validate, and use a novel approach for addressing selection when studying COVID-19 mortality risk [4,5,21]. We use the natural mortality rate from causes other than COVID-19 (non-COVID-19 NMR) as a proxy for unobserved health and thus background mortality risk, within population groups defined by age, gender, vaccination status, and other characteristics. We use a related measure, the COVID-19 excess mortality percentage (CEMP) (defined as the COVID-19 mortality rate divided by the non-COVID-19 NMR, converted to a percentage) as our principal outcome measure. This measure is available on a population-wide basis and performs well in predicting COVID-19 mortality for unvaccinated populations. In this study, we use the CEMP measure to assess RMRs for Pfizer and Moderna vaccinees versus the unvaccinated, and the relative effectiveness of each vaccine. 

## 2. Data and Methods

We study all deaths among adults age 18+ residing in Milwaukee County, Wisconsin (Milwaukee), a racially, ethnically and economically diverse county, which includes 722,000 adults, of whom 19,895 died during the COVID-19 pandemic period from 1 April 2020 through to 30 June 2022. We use death certificate data, which includes residence zip code, age at death, gender, race/ethnicity, education, income, manner of death, and text fields for cause of death and conditions leading to death. 

We use text analysis of death records to identify deaths due to COVID-19. Our text analysis identifies substantially more COVID-19 deaths than the ICD-10-based cause-of-death codes assigned by the National Center for Health Statistics (NCHS) based on the text fields, and reduces misattribution of COVID-19 deaths as non-COVID-19 deaths. See the Appendix A for further details on our text-based algorithm. Appendix A compares our COVID-19 death counts to those obtained using ICD-10 codes assigned by the NCHS. 

We study the mRNA vaccines from Moderna (mRNA1273 and Pfizer-BioNTech (BNT162b2), both of which use two initial does; we exclude persons who received the viral-vector based J&J vaccine (for whom one dose was standard), mRNA vaccinees who received only one dose, and vaccinees who received more than one type of vaccine. Standard U.S. two-dose timing was four weeks between doses for Moderna and three weeks for Pfizer. Mixing of vaccine types was uncommon (Appendix A). We treat vaccination as effective against mortality beginning 30 days after receipt, to allow for time from vaccination to full effectiveness against infection, plus the typical several-week lag from infection to death. 

We study the vaccine-available period from 1 April 2021 through to 30 June 2022. Thus, our sample period includes the first half of 2022 (1H-2022), when Omicron was the dominant COVID-19 variant. We measure VE and RMR for persons who received two or three Pfizer or Moderna doses versus the unvaccinated population, and relative RMR for Pfizer versus Moderna vaccinees. 

We define CEMP as COVID-19 deaths/non-COVID-19 natural deaths, converted to a percentage. We measure VE against death, and relative mortality risk (RMR = 1–VE) versus the unvaccinated for combinations of vaccine type, number of doses, and time period. More specifically, we define CEMP, VE versus the unvaccinated, and RMR for two-dose or three-dose vaccinees versus the unvaccinated in each time period, within a population group, as follows:$$CEMP=\frac{COVID\;deaths}{non-COVID\;natural\;deaths}$$
$$VE=\frac{\left(CEMP_{unvax}-CEMP_{vax}\right)\;}{CEMP_{unvax}}$$
$$RMR=1-VE=\frac{CEMP_{vax}\;}{CEMP_{unvax}}$$

We use similar formulas for VE and RMR to compare Pfizer to Moderna vaccinees. See the Appendix A for additional details on variable definitions. We report results for the following three time periods: pre-booster period (1 April–30 September 2021), Delta period (1 October–31 December 2021, below “4Q-2021”), and Omicron period (1 January–30 June 2022), but provide results by calendar quarters in the Appendix A. We study two-dose vaccinees in each time period, and three-dose vaccinees beginning 4Q-2021, when booster doses became available. Booster availability began in late September 2021, but was initially limited primarily to persons aged 60+. Broader booster availability began in December 2021. 

CEMP, RMR, and the ratio of CEMP for Pfizer to CEMP for Moderna (the Pfizer/Moderna ratio) are all ratios, so will be undefined if the denominator is zero. This issue did not affect our measures of CEMP and RMR versus the unvaccinated, but did prevent us from computing some Pfizer/Moderna ratios for younger persons, due to no deaths among Moderna vaccinees.

CEMP represents the odds, for a population of interest, of dying from COVID-19 versus other natural causes. The ratio of CEMPs for two groups, such as two-dose vaccinees versus unvaccinated, or Pfizer versus Moderna vaccinees, is the RMR for the two groups, and is also an odds ratio. These odds ratios can be either computed directly or obtained through logistic regression. We use both approaches. An advantage of multivariate logistic regression is that one can go beyond differences in CEMP rates between two groups, and also control for within-group differences in the regression covariates. In the regression analysis of the association between vaccine type and RMR, the predictors are vaccine type; days since most-recent dose (minus 30 days), to allow for vaccine waning; age, age^2^, zip-code-level socio-economic status (zip-SES), measured using the Graham Social Deprivation Index [22], gender, race/ethnicity (non-Hispanic White (“White”) versus other), and education (high-school or less versus college education or more). Using additional predictors was not feasible given the limited number of deaths of vaccinees.

CEMP treats non-COVID-19 natural deaths as a proxy for the health of a given group, and thus the likelihood of COVID-19 mortality if not vaccinated. We assessed the validity of this approach by studying the correlation in Wisconsin between natural mortality in April-December 2019 (pre-COVID-19) and COVID-19 mortality over the same months in 2020 (Appendix A). 

## 3. Results

### 3.1. Study Population and Evidence on Selection Effects

In the Appendix A, we provide information on the vaccination and natural mortality counts for Milwaukee adults during our sample period. Of 542,152 adult vaccinees, we excluded 94,221 because they did not receive two or three Pfizer or Moderna vaccines, and 5843 immune-compromised persons. This left a sample of 442,088 Pfizer or Moderna vaccinees, as well as 179,366 unvaccinated persons. Overall, through June 2022, around 74% of Milwaukee County adults received at least one dose, of whom 82% received two Pfizer or two Moderna doses; of these two-dose vaccinees, 57% received a third dose of the same vaccine. Among mRNA vaccine recipients, 66% received Pfizer. Only a small percentage of vaccinees received J&J or different vaccine types across doses. 

Of the 10,140 deaths of adult Milwaukee County residents during our study period, we excluded 1,605 because they involve non-natural causes (e.g., suicide, homicide and accidental death), 1064 because they received different vaccine regimens than two or three doses of Pfizer or Moderna, and 289 because they involved immune-compromised persons. This left a sample of 8250 decedents from natural causes, of whom 729 died of COVID-19. See Table 1 and Appendix A for sample selection details.

Table 1 provides summary statistics for the sample of 8250 natural deaths. There are large differences in various characteristics between the unvaccinated and vaccinated, between Pfizer and Moderna vaccinees, and between two-dose and three-dose vaccinees. Relative to two-dose recipients, the unvaccinated are younger, much less likely to be White, more likely to be male, and less educated. Relative to two-dose recipients, three-dose recipients are older, more likely to be White, and better educated. Moderna vaccinees are older than Pfizer vaccinees and more likely to be White. These differences provide initial evidence on the existence of selection effects. 

Centrally for this project, for ages 60+, two-dose Pfizer vaccinees are substantially healthier (they have lower non-COVID-19 NMR) than two-dose Moderna vaccinees, and three-dose Pfizer vaccinees are substantially healthier than three-dose Moderna vaccinees. The Pfizer vaccinees also have substantially higher CEMP levels. For younger two-dose vaccinees, age 18–59, selection effects are smaller. Three-dose Pfizer vaccinees are healthier than three-dose Moderna vaccinees, but there are no COVID-19 deaths in either group.

We confirm in the Appendix A, the existence of large selection effects within finer age groups. For all ages, vaccinees are substantially healthier (less likely to die of other natural causes) than the unvaccinated, and three-dose recipients are healthier than two-dose recipients. For ages 60+, Pfizer vaccinees are much healthier than Moderna vaccinees. For two-dose vaccinees over the full sample period, non-COVID-19 NMR for Pfizer vaccinees is 52.6% of that for Moderna. For three-dose vaccinees over the booster-available period, non-COVID-19 NMR for Pfizer vaccinees is 56.0% of that for Moderna.

### 3.2. Two-Dose RMRs and the Two-Dose Pfizer/Moderna Ratio

Table 2 reports the number of COVID-19 deaths, non-COVID-19 natural deaths, CEMP (the ratio of the two), Pfizer and Moderna RMRs versus the unvaccinated, and the Pfizer/Moderna ratio, in groups defined by age range (18–39, 40–59, 60–79, and 80+), number of doses, and vaccine type, for the following three time periods: pre-booster (April–September 2021) with Alpha and Delta as the dominant virus variants; October–December 2021, with Delta dominant but boosters available; and January–June 2022, with Omicron dominant and boosters available.

We present results by period, given evidence from other studies on vaccine waning over time [7,23,24,25,26,27,28,29], differences in severity between the Delta and Omicron variants, and potential differences in RMR between variants. Some death counts in individual cells are small, so confidence intervals for RMRs and Pfizer/Moderna ratios are wide. 

For ages 18–59, there were few vaccinee deaths and no evidence of a difference in effectiveness between vaccines. RMR versus unvaccinated persons was near zero for ages 18–39, with only one COVID-19 death among two-dose recipients. There were no COVID-19 deaths for ages 40–49 and 8 for ages 50–59 (four Pfizer and four Moderna recipients). 

For ages 60+, where most COVID-19 deaths occur, CEMP in the pre-booster period was 3.3% for two-dose Pfizer vaccinees versus 1.4% for two-dose Moderna vaccinees (Pfizer/Moderna ratio of 229%). In the Delta-and-booster period, CEMP was 12.9% for two-dose Pfizer vaccinees versus 5.4% for two-dose Moderna vaccinees (Pfizer/Moderna ratio of 240%). In the Omicron-and-booster period, CEMP was 11.2% for two-dose Pfizer vaccinees versus 4.4% for two-dose Moderna vaccinees (Pfizer/Moderna ratio of 254%). All ratios were significantly different from 100% at the 5% level or better.

For each vaccine, two-dose protection for ages 60+ waned during the study period, with RMRs for Pfizer two-dose vaccinees versus the unvaccinated rising from 34.9% in the pre-booster period to 44.1% in 4Q-2021; and 55.6% in the Omicron period. Two-dose RMRs for Moderna also rose, from 15.2% in the pre-booster period to 18.4% in 4Q-2021 and 21.8% in the Omicron period. However, the Pfizer/Moderna ratios were consistent across the three time periods. Note that we cannot separate the effects of waning over time from the changes in the dominant virus variant or the increasing likelihood of previous infection.

### 3.3. Three-Dose RMRs and the Three-Dose Pfizer/Moderna Ratio

For ages 18–59, the number of deaths among vaccinees, already small for two-dose recipients, was zero for our sample for recipients of three doses of either Pfizer or Moderna. 

For ages 60+, a booster dose offered substantial additional protection against death, with similar protection levels for Pfizer vs. Moderna vaccinees. The RMRs for booster recipients versus the unvaccinated population were 13.4% (Moderna) vs. 6.3% (Pfizer) during 4Q-2021, and 8.7% (Moderna) versus 13.1% (Pfizer) in 1H-2022 (the Pfizer-Moderna differences are not statistically significant). 

Figure 1 summarizes in graphical form the principal RMR results for ages 60+ from Table 2. It shows RMR data points by time period for two-dose Pfizer recipients, two-dose Moderna recipients, and during the booster period, for three-dose Pfizer and three-dose Moderna recipients, all versus the unvaccinated. The upward slopes over time for two-dose vaccinees confirm waning vaccine effectiveness over time. The gap between the Pfizer and Moderna lines for two-dose vaccinees shows the Moderna advantage over Pfizer. The solid lines for three dose recipients are well below the two-dose lines and thus illustrate the value of the third booster dose in reducing mortality, for both vaccines. The gap between the two-dose and three-dose lines provides a measure of the mortality risk reduction from a booster. The Figure also shows that RMRs for both vaccines are similar after three doses. The reduction in RMR from a third dose is larger for Pfizer than for Moderna; reflecting higher Pfizer two-dose RMRs but similar three-dose RMRs.

### 3.4. Multivariate Estimates

In Table 3, we use a multivariate logistic regression model to predict the Pfizer/Moderna ratio for two- and three-dose recipients, for the same sample as in Table 2. Over the full sample period, for two-dose vaccinees aged 60+, the Pfizer/Moderna ratio in Table 3 is 258% (CI = 182%,366%; *p* < 0.001). This estimate is very close to the 248% (CI = 175%,353%) estimate from the simpler comparisons in Table 2. For each subperiod, the multivariate estimates from Table 3 are again similar to those from Table 2. Thus, the additional covariates included in the multivariate model do not strongly affect the results from Table 2, in which mortality for two-dose Pfizer recipients age 60+ is over twice that for Moderna recipients.

In Table 3, for three-dose vaccinees aged 60+ over the full booster period, the Pfizer/Moderna ratio was 135% (not statistically different from 100%)**.** This is also similar to the results presented in Table 2, for which the full-booster-period estimate is 134% (CI = 63%, 283%).

Because of the small number of vaccinated decedents, especially for ages 18–59 and three-dose vaccinees, Table 3 reports results using the following limited covariates to preserve the regression degrees of freedom: gender, age, age^2^, and days since last vaccine dose (to allow for waning). However, point estimates are similar in regressions, which also control for race/ethnicity, education, and zip-SES (Appendix A).

The similarity between the simpler estimates in Table 2 and the multivariate estimates provides evidence that the CEMP denominator is effective at controlling for population health and thus for COVID-19 risk, even without controlling for additional covariates.

### 3.5. Robustness Checks

The Pfizer/Moderna ratio for ages 60+ is similar if we do not exclude the immune-compromised (Appendix A) or exclude the immune-compromised, defined more broadly than in the text (Appendix A). The results for this ratio are similar for men and women (Appendix A), and for White populations versus non-White populations (Appendix A).

## 4. Discussion

### 4.1. Prior Literature Comparing Pfizer to Moderna

Among other studies of VE against death, some study only a single vaccine type (e.g., Israeli studies of Pfizer; manufacturer-sponsored studies). Of those that study both vaccines, many report only combined results rather than vaccine-specific results [5,10,11,12,13,14,30]. Some do not report separate results for homologous versus heterologous vaccination. Among the studies of homologous vaccination that distinguish between vaccine types, some do not find substantial differences between Pfizer and Moderna. Others find differences, sometimes similar in magnitude to those reported, but do not highlight them. Only one study, limited to U.S. veterans, includes the Omicron-dominant period (only a short part of that period), or studies separately three-dose vaccinees [9].

The only other U.S. study that reports RMR using linked population-wide mortality and vaccination data is Robles-Fontan et al. (2022), who study Puerto Rico through mid-October 2021 (pre-Omicron and pre-booster). They report two-dose RMRs after 144 days (longest period considered) of 14% for Pfizer and 7% for Moderna, versus 3% and 1% soon after vaccination [31]. They thus find a Pfizer/Moderna ratio similar to ours, but their abstract states only that “Both vaccines were highly effective across all age groups.” Lytras et al. (2022) study Greece through the year-end of 2021 and find a nearly 3:1 Moderna advantage against mortality, but this result must be extracted from a supplemental figure; the text (at 5048) reports “only marginal differences between vaccines in effectiveness.” [32] Mayr et al. (2022) report a Moderna advantage in reducing hospitalization risk, and an apparent advantage for a combined ICU-or-death outcome, but the small sample size “precluded statistically significant comparisons.” [33] A study of Czechia through November 2021 reports two-dose RMR, 7–8 months after vaccination, of 17% for Pfizer vs. 12% for Moderna (2022) [34]. The review by Black and Thaw (2022) reports a Moderna advantage against mortality after waning (at least 120 days after vaccination) during the Delta-dominant period, with midpoint RMR estimates from multiple studies of 13.3% for Pfizer vs. 9.2% for Moderna [7]. Other studies find smaller differences. Islam et al. (2022) study the pre-booster period; they report an insignificant Moderna advantage for a combined hospitalization-or-death outcome during the first 90 days after vaccination [35]. Several studies of U.S. veterans find no significant Pfizer-vs-Moderna differences [8,9,36,37].

### 4.2. Two-Dose Pfizer-vs-Moderna RMRs for Ages 60+

Our analysis can help to reconcile these disparate results. We study CEMP as the principal outcome, which controls for selection effects between Pfizer and Moderna vaccinees. We also study a longer period, including the Omicron-dominant period through 30 June 2022. For ages 18–59, we find similar performance for both vaccines. In contrast, for ages 60+, we find substantially higher two-dose RMRs for Pfizer versus Moderna vaccinees. The Pfizer/Moderna ratio is at least 2:1 for ages 60+ in each of our three sample time periods.

A plausible explanation for the Pfizer-vs-Moderna differences for older people is that younger people benefit sufficiently from the boost to their immune system provided by two doses of either vaccine. Beyond some threshold level, which both vaccines achieve, the magnitude of the boost is less important. In contrast, older people may need a larger dose for full protection. This speculation would be similar to the flu vaccine, for which the recommended dose is 4x higher for ages 60+ [38]. 

This speculation is potentially testable through a clinical trial, in which different doses are provided to similar people, and antibody response over time is measured. At the same time, studying death (or even hospitalization) as an outcome seems infeasible at plausible sample sizes for such a trial. These outcomes will be rare events among study participants who were previously vaccinated, and often previously infected. The unvaccinated are typically so by choice, so will be hard to recruit for a vaccine trial, and most are also previously infected. 

### 4.3. Results for Waning and Absolute RMR versus Unvaccinated

For ages 60+, where most vaccinee deaths occur, both vaccines showed waning over time, although the evidence on waning could be confounded by changes over time in the dominant virus variants. The rise in RMR levels versus the unvaccinated is higher for Pfizer than for Moderna, but the Pfizer/Moderna ratios are consistent over our time periods.

For two-dose recipients aged 60+, we report substantially higher RMR estimates than most other studies, especially during the Omicron period. During this period, two-dose RMR versus the unvaccinated population is 23% (Moderna) and 57% (Pfizer). These estimates likely reflect a combination of waning, a higher percentage of previously infected persons in the population, who have post-infection resistance even if unvaccinated (thus, the relative gain from vaccination may be smaller), our use of CEMP to control for selection effects when measuring COVID-19 mortality risk, and perhaps changes over time in the use of non-vaccine risk mitigation measures.

### 4.4. The Value of mRNA Boosters, Especially for Pfizer

For ages 60+, a booster dose provides substantial additional reduction in RMRs for both vaccines, especially during the Omicron period. A booster dose reduced Pfizer RMR from 57% to 13%. A booster dose also reduced RMR for Moderna vaccinees, from 23% to 9% in the Omicron period. For three-dose vaccinees, we did not find significant Pfizer vs. Moderna differences in RMR. The Pfizer-versus-Moderna point estimate is 134% but with a very wide CI of (63%, 283%), due to few deaths of booster recipients. One would need a much larger sample than was available to us to assess whether there might be a significant Pfizer-versus-Moderna difference for booster recipients. The data available to us are consistent with the results of the third dose, allowing Pfizer to catch up to Moderna.

For ages 60+, our results imply a much higher booster value than prior studies, especially for Pfizer. In effect, the higher two-dose RMRs that we find increase the value of boosters, because they leave more room for boosters to reduce mortality. The large gains in RMR from a booster dose are found even though we also find higher three-dose RMRs than prior research. A UK study found 1.3% RMR for boosted versus unvaccinated populations for ages 50+ (when studies report VE, we convert it to RMR) [39]. One Israeli study reported 10% RMR for three-versus-two-doses for ages 50+ [40]; a second reports three-versus-two-dose RMR of 6.8% for ages 60+ [24]; a third reports three-versus-two-dose RMR of 19% across all ages [41]. In contrast, our results imply three-versus-two-dose RMRs of 23% for Pfizer and 39% for Moderna.

For ages 18–59, there are few deaths of two-dose vaccinees, including only one death for ages 18-49. A third dose is still valuable for ages 50–59. There was evidence for waning (higher two-dose RMR) in the Omicron period for both vaccines, and thus for the value of a booster dose for ages 50+. Zero deaths among booster recipients aged 18–59 through June 2022 is a striking result, which suggests value in a third dose, separated in time from the initial two doses. At the same time, at least through mid-2022, persons aged 18–49 are already well protected against death after two doses.

## 5. Limitations

This study has important limitations. We study only mortality. The results could be different for other measures of severe disease, such as hospitalization or admission to the ICU, or for risk of long COVID. However, prior work has found that relative Pfizer vs. Moderna VE against hospitalization is similar to VE against mortality [4,6].

We have data only for Milwaukee County. Milwaukee County is racially, ethnically, and economically diverse, but may not be representative of other areas. However, the vaccination patterns in Milwaukee County (Appendix A) are broadly similar to those observed nationally. Also, although we are the first to highlight the Moderna advantage over Pfizer for two-dose vaccinees, we are not the first to find a substantial Moderna advantage; see also [7,31,32,33,34].

Third, we rely on non-COVID-19 natural mortality as a surrogate for the underlying risk of COVID-19 death. This measure is theoretically attractive. It is conceptually similar to a “*p*-value”, which is sometimes computed for all-cause excess mortality (P = excess mortality as a percentage of expected mortality) [42,43]. We confirmed that non-COVID-19 natural mortality rates in 2019, prior to COVID-19, strongly predict COVID-19 mortality rates in 2020, when COVID-19 vaccines were not available, for population groups defined by age, gender, and race/ethnicity (Appendix A; Pearson correlation coefficient = 0.94).

Fourth, we lack data on prior COVID-19 infection. Especially in the Omicron era, when many people were already infected, the comparison of unvaccinated to vaccinated persons could be affected by differences in the proportion of persons in each group who have some natural resistance, due to prior infection. Moreover, RMRs versus the unvaccinated could be affected by prior infection. However, unless prior COVID-19 infection is associated with the choice of vaccine, estimates of the Pfizer/ Moderna ratio should still be unbiased.

Fifth, some deaths due primarily to COVID-19 may be coded as non-COVID-19 natural deaths. Moreover, COVID-19 infection predicts higher near-term mortality from other causes [44,45]. However, we coded COVID-19 deaths based on text fields in death certificates to reduce miscoding (Appendix A). For our sample, the rate of non-COVID-19 natural mortality during the study period was similar to that predicted by extrapolating natural mortality rates from the pre-pandemic period (Appendix A). Any miscoding of COVID-19 as non-COVID-19 natural deaths will reduce CEMP estimates, but we have no reason to expect this to produce bias in the Pfizer/Moderna ratio.

Finally, our assessment of underlying health does not control for behavioral differences between the vaccinated and unvaccinated. However, we have no reason to expect behavioral differences between the people receiving Moderna versus Pfizer vaccines.

## 6. Conclusions

We provide several main results that can inform clinical practice, public health guidance, and the development of COVID-19 vaccines. First, we provide evidence that when comparing the effectiveness of different vaccines, one must address selection effects for which people receive which vaccine. Second, we report evidence that one COVID-19 vaccine policy does not fit all recipients. After controlling for selection effects, we find that two doses of the Moderna vaccine are strongly preferable to two Pfizer doses for ages 60+, at least until one receives a booster dose. RMRs for two-dose Pfizer vaccinees aged 60+ are more than double those for Moderna recipients. This suggests that the general advice for older persons to obtain a booster dose should be reinforced for Pfizer recipients, who are at higher risk of mortality without a booster. However, the Pfizer versus Moderna difference is insignificant after a booster dose. The Pfizer versus Moderna difference is also insignificant for persons aged 18–59, after either two or three doses.

Second, RMR estimates for persons aged 60+ are much higher, and therefore VE estimates are much lower than the estimates from prior research, much of which did not effectively address the selection effects for those who are vaccinated and boosted. Our RMR estimates underscore the importance of a booster dose in this age range, and the lives that could have been saved, and could still be saved, by higher booster take-up.

Conversely, younger persons are well protected by two-doses of either vaccine. For this population group, the lower Pfizer dose may have a lower risk of side effects, especially myocarditis and pericarditis, which are important side effects for young men, with higher risk for Moderna than for Pfizer. The higher Moderna dose provides a plausible, although as yet unproven, explanation for this increased risk.

Current U.S., EU, and UK public health messaging does not distinguish between vaccines, nor, for the first booster, between different ages. Instead, public health agencies have promoted boosters for all. Our evidence suggests the need for more nuanced guidance that can reflect differences in the response to vaccination by recipient age, such as those our research has highlighted. Differences based prior infection status are also plausible, although we lacked the data to study them.

Our results suggest that vaccine manufacturers need to investigate how the immune response to COVID-19 vaccination, and waning of that response over time, varies with age and prior infection.

It will be important to monitor differences in effectiveness for both two-dose and three-dose vaccinees over time to determine if waning of booster protection differs between the two vaccines, to do so by age range, and to assess whether our results will carry over to new variants, when and if important new variants emerge. It will also be important to assess relative effectiveness against hospitalization, as we studied only mortality.

## Figures and Tables

**Figure 1 vaccines-11-00971-f001:**
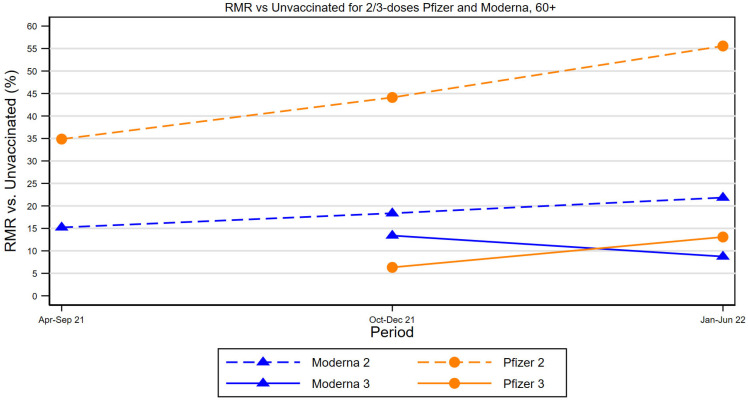
Summary of Two-Dose and Three-Dose RMRs for Pfizer and Moderna, Ages 60+. Figure summarizes RMRs from Table 2, for two-dose and three-dose Pfizer and Moderna vaccinees aged 60+, relative to the unvaccinated population, for the indicated times periods. The Figure shows increased two-dose RMRs over time, especially during the Omicron period, higher RMRs for Pfizer than for Moderna two-dose recipients, the reduction in RMRs for both vaccines from a booster dose, and similar RMRs for both vaccines after a booster dose.

**Table 1 vaccines-11-00971-t001:** Sample Summary Statistics.

	Vaccination Status									
Variable	Unvaccinated		Moderna 2 Doses		Pfizer 2 Doses		Moderna 3 Doses		Pfizer 3 Doses	
Pop. at June 1, 2022	179,366		64,186		124,879		95,946		157,077	
Monthly average pop.	250,078		107,854		179,932		40,232		68,288	
Number of deaths	4535		2173		1621		851		675	
Mean age at death	65		78		71		82		76	
Age at death *n*(%)										
18–39	614	13.5%	47	2.2%	77	4.8%	3	0.4%	12	1.8%
40–59,	977	21.5%	167	7.7%	257	15.9%	45	5.3%	60	8.9%
60–79	1752	38.6%	795	36.6%	740	45.7%	271	31.8%	301	44.6%
80 +	1192	26.3%	1164	53.6%	547	33.7%	532	62.5%	302	44.7%
Female	1964	43.3%	1226	56.4%	786	48.5%	524	61.6%	301	44.6%
Non-White	2180	48.1%	524	24.1%	604	37.3%	147	17.3%	184	27.3%
High school and below	3122	68.8%	1249	57.5%	977	60.3%	463	54.4%	380	56.3%
Mean Graham SDI	78.3		67.1		73.7		63.8		67.4	
**Age 18–59**										
COVID-19 deaths	147	18.2%	4	3.3%	6	3.0%	0	0.0%	0	0.0%
Non-COVID-19 Natural deaths	660	81.8%	116	96.7%	197	97.0%	38	100.0%	45	100.0%
CEMP		**22.27%**		**3.45%**		**3.05%**		**0.00%**		**0.00%**
Non-COVID-19 NMR		**0.31%**		**0.17%**		**0.15%**		**0.19%**		**0.11%**
**Age 60+**										
COVID-19 deaths	397	14.5%	58	3.2%	89	7.6%	14	1.9%	14	2.5%
Non-COVID-19 natural deaths	2341	85.5%	1759	96.8%	1088	92.4%	731	98.1%	546	97.5%
CEMP		**16.96%**		**3.30%**		**8.18%**		**1.92%**		**2.56%**
Non-COVID-19 NMR		**5.95%**		**4.52%**		**2.31%**		**3.60%**		**1.88%**

Notes: Table shows summary statistics for study sample. First two rows provide vaccination information for 621,454 adult residents of Milwaukee County, Wisconsin, who received two or three doses of either the Pfizer or the Moderna COVID-19 vaccines, or were unvaccinated, excluding immune-compromised persons. Remaining rows provide information on 8250 decedents who died of natural causes over sample period, from 1 April 2021 through 30 June 2022. Higher SDI indicates lower zip SES. Monthly average population by number of vaccine doses is average of beginning of month populations over 1 April 2021–1 June 2022. Non-COVID-19 NMRs are based on monthly average populations.

**Table 2 vaccines-11-00971-t002:** Relative Mortality Risks and Pfizer/Moderna Ratio by Age Group and Time Period.

		April–September 2021			October–December 2021					January–June 2022				
Ages	Death	Unvax	M2	P2	Unvax	M2	P2	M3	P3	Unvax	M2	P2	M3	P3
18–39	COVID-19 deaths	9	0	0	16	0	0	0	0	4	0	1	0	0
	Non-COVID-19 natural deaths	63	5	5	26	5	6	0	0	32	4	13	1	2
	CEMP	14.3%	0.0%	0.0%	61.5%	0.0%	0.0%	NA	NA	12.5%	0.0%	7.7%	0.0%	0.0%
	RMR to unvax		0.0%	0.0%		0.0%	0.0%	NA	NA		0.0%	61.5%	0.0%	0.0%
	Pfizer/ Moderna		NA			NA			NA		NA		NA	
40–59	COVID-19 deaths	38	0	1	57	1	1	0	0	23	3	3	0	0
	Non-COVID-19 natural deaths	260	28	52	102	31	63	1	2	177	43	58	36	41
	CEMP	14.6%	0.0%	1.9%	55.9%	3.2%	1.6%	0.0%	0.0%	13.0%	7.0%	5.2%	0.0%	0.0%
	RMR to unvax		0.0%	13.2%		5.8%	2.8%	0.0%	0.0%		53.7%	39.8%	0.0%	0.0%
	Pfizer/ Moderna		NA			49.2%		NA			74.1%		NA	
60–79	COVID-19 deaths	75	4	9	95	13	17	1	0	90	7	18	3	6
	Non-COVID-19 natural deaths	664	294	269	266	202	175	15	28	416	219	185	234	241
	CEMP	11.3%	1.4%	3.3%	35.7%	6.4%	9.7%	6.7%	0.0%	21.6%	3.2%	9.7%	1.3%	2.5%
	RMR to unvax		12.0%	29.6%		18.0%	27.2%	18.7%	0.0%		14.8%	45.0%	5.9%	11.5%
	Pfizer/Moderna		245.9%			150.9%		0.0%			**304.4% ***		194.2%	
80+	COVID-19 deaths	32	8	7	49	15	22	1	1	56	11	16	9	7
	Non-COVID-19 natural deaths	462	536	214	226	319	127	36	26	307	189	118	446	251
	CEMP	6.9%	1.5%	3.3%	21.7%	4.7%	17.3%	2.8%	3.8%	18.2%	5.8%	13.6%	2.0%	2.8%
	RMR to unvax		21.5%	47.2%		21.7%	79.9%	12.8%	17.7%		31.9%	74.3%	11.1%	15.3%
	Pfizer/ Moderna		219.2%			**368.4% *****		138.5%			**233.0% ***		138.2%	
18–59	COVID-19 deaths	47	0	1	73	1	1	0	0	27	3	4	0	0
	Non-COVID-19 natural deaths	323	33	57	128	36	69	1	2	209	47	71	37	43
	CEMP	14.6%	0.0%	1.8%	57.0%	2.8%	1.4%	0.0%	0.0%	12.9%	6.4%	5.6%	0.0%	0.0%
	RMR to unvax		0.0%	12.1%		4.9%	2.5%	0.0%	0.0%		49.4%	43.6%	0.0%	0.0%
	Pfizer/ Moderna		NA			52.2%		NA			88.3%		NA	
60+	COVID1-9 deaths	107	12	16	144	28	39	2	1	146	18	34	12	13
	Non-COVID-19 natural deaths	1126	830	483	492	521	302	51	54	723	408	303	680	492
	CEMP	9.5%	1.4%	3.3%	29.3%	5.4%	12.9%	3.9%	1.9%	20.2%	4.4%	11.2%	1.8%	2.6%
	RMR to unvax		15.2%	34.9%		18.4%	44.1%	13.4%	6.3%		21.8%	55.6%	8.7%	13.1%
	Pfizer/ Moderna		**229.1% ***			**240.3% *****		47.2%			**254.3% ****		149.7%	

Notes: Sample is same as Table 1. Table shows COVID-19 deaths, natural non-COVID-19 deaths, COVID-19 excess mortality percentage (CEMP), RMR relative to the unvaccinated population for vaccinees with indicated vaccine types (Pfizer = P; Moderna = M), and Pfizer/Moderna ratio of RMRs, by number of doses. Vaccine doses are considered effective 14 days after receipt. RMR for a comparison of two groups is the ratio of CEMP for group 1 to CEMP for group 2. Sample is adult decedents in Milwaukee County, Wisconsin, excluding immune-compromised persons, who were unvaccinated or received two or three Pfizer or Moderna doses. Due to the nature of the sample, CEMP ratios and RMRs are effectively weighted by natural mortality rates. *, **, *** indicates *p* < 0.05, 0.01, and 0.001, respectively; significant results are (at *p* < 0.05 or better) in boldface.

**Table 3 vaccines-11-00971-t003:** Comparative RMR of Pfizer vs. Moderna from Multivariate Logistic Model.

		2-Dose Recipients			3-Dose Recipients		
Age in Years	Period	P/M Ratio	*p*-Value	95 CI	P/M Ratio	*p*-Value	95 CI
18–59	April–September 2021	No COVID-19 deaths		NA			
	October–December 2021	130.6%	0.887	(3.3%, 5179.4%)	No deaths	NA	NA
	January–June 2022	97.2%	0.973	(19.3%, 489.8%)	No deaths	NA	NA
	January 2021–June 2022	103.3%	0.961	(28.5%, 374.4%)	No deaths	NA	NA
60+	April–September 2021	**285.4%**	**0.010**	**(128.4%, 634.1%)**			
	October–December 2021	**254.2%**	**0.001**	**(149.5%, 432.2%)**	36.1%	0.474	(2.2%, 585.8%)
	January–June 2022	**238.9%**	**0.005**	**(130.6%, 437.1%)**	152.6%	0.324	(65.9%, 353.5%)
	January 2021–June 2022	**257.8%**	**<0.001**	**(181.6%, 366.1%)**	134.9%	0.454	(61.6%, 295.5%)

Notes: Table shows odds ratios from logistic estimation of COVID-19 mortality for samples of persons in Milwaukee County, aged 18–59 or aged 60+, who died of natural causes and received 2 or 3 doses of either Pfizer or Moderna over indicated periods. Odds ratios are for Pfizer vaccinee mortality relative to Moderna vaccinees (P/M ratio), from a logistic model of Prob (COVID-19 Death) = f (received Pfizer (Moderna is baseline), with controls for age, age^2^, gender, and (days since last vaccine dose, minus 30 days). Sample is same as Table 1. Significant results (at *p* < 0.05 or better) in boldface.

## Data Availability

The linked mortality and vaccination data on which this study relies were obtained under a data use agreement with the Wisconsin Department of Health Services, and cannot be publicly shared.

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
