# Peer review of "Selection Effects and COVID-19 Mortality Risk after Pfizer vs. Moderna Vaccination: Evidence from Linked Mortality and Vaccination Records"

_vaccines, 2023, doi:10.3390/vaccines11050971_

Round 1

Reviewer 1 Report

Dear authors, 

thanks a lot for the opportunity to serve as a reviewer of your very interesting article submitted to Vaccines for potential consideration and publication. It is a very scientifically sound, well-written, and easy-to-follow paper on an important topic. The methodology enables the advancement of knowledge in a field in which vaccine hesitancy may compromise the efforts of immunization mass campaigns and, as such, has major, practical public health implications in the field of vaccinations and healthcare policies.  

The first paragraph of the introduction is entirely without any scholarly reference. Please, amend it accordingly. 

The methods section should be expanded: for instance, authors could show some formulas and provide more details about the methodology, which is the core of their investigation. 

Can authors draw some geo-maps?

Author Response

Selection Effects and COVID-19 Mortality Risk After Pfizer vs. Moderna Vaccination: Evidence from Linked Mortality and Vaccination Records

Vaccines Editor and Reviewers (25 April 2023)

Editor’s Reply

[Cover email subject line was: Vaccines] Manuscript ID: vaccines-2338972 - Minor Revisions (Due 2 May 2023)

(I) Please revise your manuscript according to the referees’ comments and upload the revised file within 5 days.

(II) Please use the version of your manuscript found at the above link for your revisions. 

(III) Please check that all references are relevant to the contents of the manuscript.

(IV) Any revisions made to the manuscript should be marked up using the “Track Changes” function if you are using MS Word/LaTeX, such that changes can be easily viewed by the editors and reviewers.

Reply.  We uploaded both marked and clean versions.  In the marked version, there are some instances where text was moved; these are not marked.

(V) Please provide a short cover letter detailing your changes for the editors’ and referees’ approval.

Reply.  We uploaded a point by point response to the referee comments.

Please do not hesitate to contact us if you have any questions regarding the revision of your manuscript or if you need more time. We look forward to

hearing from you soon.

Reviewer 1:

thanks a lot for the opportunity to serve as a reviewer of your very interesting article submitted to Vaccines for potential consideration and publication. It is a very scientifically sound, well-written, and easy-to-follow paper on an important topic. The methodology enables the advancement of knowledge in a field in which vaccine hesitancy may compromise the efforts of immunization mass campaigns and, as such, has major, practical public health implications in the field of vaccinations and healthcare policies. 

The first paragraph of the introduction is entirely without any scholarly reference. Please, amend it accordingly.

Reply:  We revised this paragraph was amended to add references for the statement that conventional wisdom treats most vaccines as similar, and also to refer forward to the literature review in the discussion section.

The methods section should be expanded: for instance, authors could show some formulas and provide more details about the methodology, which is the core of their investigation.

Reply:  We now include the main formulas in the text; previously they were only in the Appendix.  We also expanded the methods section more generally.  There is still a judgment call on what to include in the text, and what to keep only in the Appendix.

Can authors draw some geo-maps?

Reply:  We are studying a relatively compact geographic area, Milwaukee County, Wisconsin, so it is not clear what one could usefully do.  One could potentially show the proportion of persons receiving Pfizer versus Moderna by 5-digit zip code, but the number of deaths among vaccinated people in each zip code is limited.  Among two-dose vaccinees age 60+, we have 147 COVID-19 deaths (we have 89 Pfizer and 58 Moderna deaths, across 38 zip codes), and the zip codes also vary in their population characteristics.  In the end, we were unable to develop useful mapping ideas.

Reviewer 2 Report

It  is  good  study  involved  : ((Effects and COVID-19 Mortality Risk After Pfizer vs. Moderna Vaccination:  )) but it  needs  Minor  corrections :

1-  There  is  no  compared  between  results in  present  study  and  previously  studies  to  improve  current  results.

2- There  are  no  Clear  conclusions

3- There  is no  explanation  of  results  in  Table (3)

4-   There  are  references  as  self  citation in references  ( 20  and 21 )  replace   by other 

5- replaces   reference  No. (30)  by other  more  updated 

6- Also  figure (1)  needs  clarification 

7- I accepted  paper  after  Minor  corrections  

Author Response

Selection Effects and COVID-19 Mortality Risk After Pfizer vs. Moderna Vaccination: Evidence from Linked Mortality and Vaccination Records

Reviewer 2

It is good study involved (Effects and COVID-19 Mortality Risk After Pfizer vs. Moderna Vaccination:  )) but it  needs Minor  corrections :

1-  There is no compared between results in present study and previously studies to improve current results.

Reply.  We discuss prior literature in the discussion section.  We revised the introduction to refer forward to that discussion.

2- There are no Clear conclusions

Reply.  We extensively rewrote the Conclusions to be more forceful in our recommendations, within the limits of what our data will support.

3- There is no explanation of results in Table (3)

Reply.  We added a more complete explanation of the multivariate results in Table 3 and how they compare to the simpler results in Table 2.

4- There are references as self citation in references (20 and 21) replace by other

Reply.  With respect, we believe that the citations to reference 4 (a systematic review) and 20 (our own prior study in this journal) are appropriate.  However, we added additional citations to other studies which discuss waning.

5- replaces reference No. (30) by other more updated

Reply.  We updated our review of the literature on the effectiveness of booster doses, but did not find additional sources that warranted discussion in the text.  There are a number of new studies of booster doses, but they either do not distinguish between vaccine types, do not distinguish between homologous and heterologous boosting, do not distinguish between first and second booster dose, or have a combination of these problems. 

6- Also figure (1) needs clarification

Reply. We expanded the discussion of this Figure in the text, and also expanded the Figure heading.

7- I accepted paper after Minor corrections

Reviewer 3 Report

Review Report
I have read and reviewed the paper “Selection Effects and COVID-19 Mortality Risk After Pfizer vs. Moderna Vaccination: Evidence from Linked Mortality and Vaccination Records” very carefully and found that the results of the paper are looking interesting and mathematically correct. The novelty of the results is also good and the presentation of the paper is suitable. I believe this paper’s results are interesting in both mathematics and applications. I recommend this paper for publication in this journal but after some corrections. My suggestions are given as follows:
1. There are some grammatical faults scattered throughout the manuscript. It has to be fixed.
2. The abstract should be improved and extended so that it can reflect the overall content of the paper.
3. The Introduction should make a compelling case for why the study is useful and clearly state its novelty or originality.
4. The introduction should include a recent literature review regarding the Covid-19 disease. I suggest the following recent works to the authors:
i. Modeling and analysis of COVID-19 epidemics with treatment in fractional derivatives using real data from Pakistan. European Physical Journal Plus, 2020, 135(10):795.
ii. Modeling the effects of the contaminated environments on COVID-19 transmission in India. Results in Physics 29:104774, 2021.
iii. Modeling the transmission dynamics of COVID-19 pandemic in Caputo-type fractional derivative. Journal of Multiscale Modelling 12(3):2150006, 2021.
5. The advantages of considering the proposed study over the existing literature should be included in the conclusion part.
6. In the conclusion part, the significant outcome of the study needs to be mentioned.
The paper can be accepted for publication after the above-suggested revisions.

Author Response

Selection Effects and COVID-19 Mortality Risk After Pfizer vs. Moderna Vaccination: Evidence from Linked Mortality and Vaccination Records

Reviewer 3

I have read and reviewed the paper “Selection Effects and COVID-19 Mortality Risk After Pfizer vs. Moderna Vaccination: Evidence from Linked Mortality and Vaccination Records” very carefully and found that the results of the paper are looking interesting and mathematically correct. The novelty of the results is also good and the presentation of the paper is suitable. I believe this paper’s results are interesting in both mathematics and applications. I recommend this paper for publication in this journal but after some corrections. My suggestions are given as follows:

  1. There are some grammatical faults scattered throughout the manuscript. It has to be fixed.

Reply.  We reread the draft, and made revisions.

  1. The abstract should be improved and extended so that it can reflect the overall content of the paper.

Reply.  We revised the abstract, but the journal limit for abstract length is 200 words.  The abstract was already over 300 words, and the editor has allowed it to exceed 200 words, but we had no room to expand further. 

  1. The Introduction should make a compelling case for why the study is useful and clearly state its novelty or originality.

Reply.  We said a bit more in the Introduction, but were limited by medical journal style, in which the Introduction is typically short, neutrally phrased, and in contrast to other disciplines, does not discuss study results.

  1. The introduction should include a recent literature review regarding the Covid-19 disease. I suggest the following recent works to the authors:
  2. Modeling and analysis of COVID-19 epidemics with treatment in fractional derivatives using real data from Pakistan. European Physical Journal Plus, 2020, 135(10):795.
  3. Modeling the effects of the contaminated environments on COVID-19 transmission in India. Results in Physics 29:104774, 2021.

iii. Modeling the transmission dynamics of COVID-19 pandemic in Caputo-type fractional derivative. Journal of Multiscale Modelling 12(3):2150006, 2021.

Reply.  We revised and updated the literature review, but did not include these sources.  With respect, we do not see important overlap between these studies of transmission dynamics and our project, which focuses on relative vaccine effectiveness. 

We kept the literature review in the Discussion section, which is more consistent with medical journal style, but now refer forward to it in the Introduction.

  1. The advantages of considering the proposed study over the existing literature should be included in the conclusion part.

Reply.  We extensively rewrote and expanded the conclusion.

  1. In the conclusion part, the significant outcome of the study needs to be mentioned.

Reply.  We believe that the revised conclusion meets this goal.

The paper can be accepted for publication after the above-suggested revisions.